# Using Subcritical Water to Obtain Polyphenol-Rich Extracts with Antimicrobial Properties

**DOI:** 10.3390/antibiotics13040334

**Published:** 2024-04-05

**Authors:** Tjaša Žagar, Rok Frlan, Nina Kočevar Glavač

**Affiliations:** 1Department of Pharmaceutical Biology, Faculty of Pharmacy, University of Ljubljana, Aškerčeva 7, 1000 Ljubljana, Slovenia; tz6991@student.uni-lj.si; 2Department of Pharmaceutical Chemistry, Faculty of Pharmacy, University of Ljubljana, Aškerčeva 7, 1000 Ljubljana, Slovenia; rok.frlan@ffa.uni-lj.si

**Keywords:** subcritical water extraction, polyphenols, phenolic acids, flavonoids, antimicrobial activity

## Abstract

The use of green extraction methods that meet the criteria of sustainable and environmentally friendly technologies has been increasing in recent decades due to their many benefits. In this respect, extracts obtained using subcritical water are also gaining increased attention because of their potential antioxidant and antimicrobial properties. Their antimicrobial activity is mainly due to the presence of various polyphenolic compounds. Although the exact mechanism of the antibacterial action of polyphenolic compounds has not yet been fully investigated and described, polyphenols are known to affect the bacterial cell at several cellular levels; among other things, they cause changes and ruptures in the cell membranes of the bacterial cell, affect the inactivation of bacterial enzymes and damage bacterial DNA. The difference in the strength of the antimicrobial activity of the extracts is most likely a result of differences in their lipophilicity and in the number and position of hydroxyl groups and double bonds in the chemical structure of polyphenols. By changing the extraction conditions, especially the temperature, during subcritical water extraction, we affect the solubility of the compounds we want to extract. In general, as the temperature increases, the solubility of polyphenolic compounds also increases, and the reduction of the surface tension of subcritical water at higher temperatures also enables faster dissolution of polyphenolic compounds. Different bacterial strains have different sensitivity to different extracts. However, extracts obtained with subcritical water extraction demonstrate strong antimicrobial activity compared to extracts obtained with conventional methods.

## 1. Introduction

In recent decades, modern, green extraction processes have been developed and implemented, which are based on the use of alternative solvents that are environmentally acceptable and ensure the production of safe and effective extracts or products, and are aimed at reducing energy consumption. These extraction approaches include modified extraction methods such as ultrasound-assisted (UAE), microwave-assisted (MAE), enzymatic-assisted (EAE) and pulsed electric field-assisted (PEF) extraction, supercritical fluid extraction (SFE), subcritical water extraction (SWE) and deep eutectic extraction (DEE) [1]. In this context, extracts obtained with subcritical water are also gaining increasing attention because of their potential antimicrobial properties. The antimicrobial activity is mainly due to the presence of various phenolic compounds [2,3]. Antimicrobial resistance is an increasing problem. In this context, the use of environmentally friendly extraction technologies presents an important field of research. Despite this, there is relatively little data available on the use of subcritical water to obtain extracts with antimicrobial activity. The purpose of this review was to analyze the currently available articles on the use of subcritical water as a solvent to obtain extracts with antimicrobial activity compared to other extraction methods and to conclude how used extraction approach influences the extract’s composition and its antimicrobial properties. 

We searched the PubMed database for studies published in the last 10 years on our main topic. The keywords were »subcritical water extraction« and »antimicrobial activity«. A total of nine studies were found in the database, of which four studies were relevant to our review. To find further studies, we also used the Google Scholar database. The keywords were »subcritical water extraction« and »antimicrobial activity«. After selecting the reviewed articles, a total of 714 studies were found. We added additional keywords (»phenolic compounds«, »MIC« and »TPC«) and found 48 studies, of which six studies were relevant to our review. Finally, we collected the results of subcritical water extraction from 13 different plants. The aspects of extraction yield, phenolic compound content and antimicrobial activity are discussed in relation to the extraction parameters, as well as the structure–activity relationship (SAR).

### 1.1. Subcritical Water Extraction

A characteristic feature of SWE is that the extraction is carried out at temperatures between 100 and 374 °C and at a sufficient pressure in the range of up to 220 bar, at which water is in a liquid state. Subcritical water penetrates the plant material: it interrupts the interactions between the solutes and the plant matrix, as well as the interactions between the solutes. This results in desorption processes of solutes from plant matrix and their dissolution in subcritical water. Under normal conditions, the dielectric constant (ε) of water is around 80, which makes water a suitable solvent for the extraction of polar compounds. With increasing temperature, the dielectric constant, density, viscosity and surface tension of subcritical water decrease, while its diffusivity increases. As a result of these changes, the dielectric constant of water approaches the dielectric constants of organic solvents such as methanol (ε = 33), ethanol (ε = 24) and acetone (ε = 21). Therefore, water becomes a good solvent for less polar compounds, whose solubility in water is low under normal conditions [4,5,6,7].

SWE can be performed in dynamic or static mode. In dynamic mode, fresh subcritical water flows continuously through the extraction cell with the plant material, which enables higher extraction yields. In static mode, the plant material in the extraction cell is in contact with subcritical water for a certain period of time. There is no continuous flow of fresh solvent. The equilibrium between the solute and the solvent is established faster and the extraction yields are lower [6,7]. A schematic diagram of the SWE system is presented in Figure 1.

The general advantages of SWE include shorter extraction time, lower extraction solvent consumption and chemically more complex extracts, as subcritical water allows the extraction of both polar and less polar compounds. In terms of analytics, the SWE process is highly compatible with various chromatographic methods, as water does not interfere with most photodetection methods. One of the main disadvantages of SWE is the use of high temperatures. This is because performing the extraction at high temperatures can lead to thermal decomposition of the compounds. In addition, extraction is also less selective at higher temperatures due to the increased solubility of many components of plant matrix. Since subcritical water is present in liquid form, the result of the extraction is a water extract. Additional processes such as evaporation, precipitation or dehydration are required to remove the water from extract solutions. In addition, water is more reactive and corrosive at subcritical conditions than water at room conditions, which requires more demanding maintenance and cleaning of the equipment [6,7,8,9].

### 1.2. Phenolic Compounds 

Plants biosynthesise many aromatic compounds that protect them from herbivores, pathogenic microorganisms, mechanical damage and oxidative stress. Among these, phenolic compounds are one of the most important and numerous [10,11]. Phenolic compounds are also an important part of our diet. The daily intake is about 1 g and they are found in fruit; vegetables; beverages; spices; and herbs, nuts and seeds [12,13,14]. Phenolic compounds are found in various parts of plants; their content is usually higher in the aerial parts of the plant [15]. 

The basic structure of phenolic compounds consists of at least one aromatic benzene ring to which one or more hydroxyl groups are attached; in the case of the latter, they are referred to as polyphenols [10]. Phenolic compounds are chemically reactive mild acidic compounds that are prone to oxidation [16]. In plants, hydroxyl groups of the phenolic compounds are most commonly bound to sugars (D-glucose, L-rhamnose, galactose, arabinose, glucorhamnose) [15,17,18]. 

Phenolic compounds are categorized into different classes and subclasses based on their chemical structure [15]. The most commonly used classification of phenolic compounds comprises five main classes: phenolic acids, flavonoids, stilbenes, lignans and tannins [18,19].

Phenolic acids are hydroxybenzoic or hydroxycinnamic acids which have a carboxyl group either directly or via an additional chain of two carbon atoms attached to the benzene ring [20]. Hydroxybenzoic acids have a C6-C1 structure and are the simplest naturally occurring phenolic acids [19]. The most important hydroxybenzoic acids include gallic acid (3,4,5-trihydroxybenzoic acid); p-hydroxybenzoic acid, protocatechuic acid (3,4-dihydroxybenzoic acid); vanillic acid (4-hydroxy-3-methoxybenzoic acid); and syringic acid (4-hydroxy-3,5-dimethoxybenzoic acid). Conversely, hydroxycinnamic acids have a C6-C3 structure. The most common hydroxycinnamic acids include ferulic acid (4-hydroxy-3-methoxycinnamic acid); caffeic acid (3,4-dihydroxycinnamic acid); p-coumaric acid (4-hydroxycinnamic acid); and sinapic acid (3,5-dimethoxy-4-hydroxycinammic acid) [14,17,21]. 

Flavonoids are a large group of phenolic compounds with the basic structure of flavone (C6-C3-C6) or 2-phenylbenzopyran (Figure 2), which consists of 15 carbon atoms [16,20,21]. The structure of flavonoids consists of two aromatic rings (ring A and ring B) connected to three carbon atoms and forming an oxygen-containing heterocycle (ring C) [12,22,23]. 

Flavonoids can be categorized into different subclasses based on the variations in the C ring [12,22]. The most common subclasses of flavonoids are flavonols, flavones, isoflavones, flavanones, flavanols, chalcones and anthocyanidins [18,20,24].
Flavonols (Figure 3) have the basic structure of a flavone and a hydroxyl group at position 3. The most important representatives of flavonols are myricetin, kaempferol and quercetin [15,20,22,23]. Flavones (Figure 2) differ from flavonols in that they do not have a hydroxyl group at position 3. Luteolin and apigenin are the most common flavones [22,23]. Isoflavones (Figure 4) differ from other flavonoids in the position of ring B. In isoflavones, a phenyl ring is attached to position 3. The most common representatives of isoflavones are genistein and daidzein [22].Flavanones (Figure 5) differ from flavones by the absence of the double bond between positions 2 and 3. Naringenin and hesperitin are the representatives of flavanones [22,23]. Flavanols (flavan-3-ol) (Figure 6) have a similar structure to flavonols but lack the double bond between positions 2 and 3 and have no oxo group at position 4. The most common flavanols are catechin and epicatechin [12,22,23]. Chalcones (Figure 7), such as flavokawin and cardamonin contain an open C ring [22].The basic structure of anthocyanidins (Figure 8) is the flavylium ion. Most anthocyanidins are derivates of the 3,5,7-trihydroxyflavylium ion [22].

Stilbenes are plant stress products. They have a C6-C2-C6 structure in which two phenyl rings are linked by ethylene bridge [14,25]. They are formed as a response of the plant to external stress. The most common stilbenes are resveratrol, pterostilbene and piceatannol [14].

Lignans are dimeric compounds consisting of two phenylpropanoid units [25]. The most common lignans include secoisolariciresinol, lariciresinol, pinoresinol and matairesinol [19,25].

Tannins are polymeric phenolic compounds with a medium to high molecular weight [22]. They can be divided into two major groups: condensed tannins (proanthocyanidins), which are oligomers or polymers of flavan-3-ol units linked by a C-C bond and hydrolyzable tannins, which are polyesters of a sugar unit and a phenolic acid [12,14,26]. If the sugar unit is linked to gallic acid, they are called gallotannins and if the acid component is hexahydroxydiphenic acid, they are called ellagitannins. The important hydrolyzable tannins include tannic acid, theogallin or punicalagin [18,22]. Tannins form insoluble complexes with proteins or metal ions, which leads to their precipitation [15,22].

#### 1.2.1. Biosynthesis of Phenolic Compounds

Phenolic compounds are mainly produced via two main metabolic pathways: the shikimate and aceto-malonate pathway [13,15]. Condensation of phosphoenolpyruvate and erythrose-4-phosphate produces 3-deoxy-D-arabinohexulose-7-phosphate. Through a series of enzyme-controlled processes 3-deoxy-D-arabinohexulose-7-phosphate is converted into shikimic acid [15]. After further conversions, various hydroxybenzoic acids, such as p-hydroxybenzoic acid, protocatechuic acid and gallic acid are formed. Shikimic acid is the precursor in the biosynthesis of aromatic amino acids (L-phenylalanine, L-tyrosine and L-tryptophan) [15,25,26]. L-phenylalanine ammonia-lyase (PAL) plays an important role in this metabolic pathway, as it enables the deamination of L-phenylalanine and the formation of *trans*-cinnamic acid [16,25,27]. Hydroxybenzoic acids are synthesized directly through the shikimic pathway and are produced even when PAL is not active [25]. The formation of *trans*-cinnamic acid represents the beginning of the phenylpropanoid pathway involved in the biosynthesis of hydroxycinnamic acids. A series of enzyme-driven processes leads to the formation of various hydroxycinnamic acids, such as p-coumaric acid, caffeic acid, ferulic acid and sinapic acid [15,25,26]. Cinnamate-4-hydroxylase (C4H) catalyses the hydroxylation of *tran*s-cinnamic acid and the formation of p-coumaric acid [15,25,27]. p-coumaric acid is then activated to p-coumaroyl-CoA by the action of 4-coumaroyl-CoA ligase (4CL) [15,25]. The phenylpropanoid pathway combines with the aceto-malonate pathway, which leads to the synthesis of flavonoids. The B ring in the flavonoid-structure in formed via the shikimate pathway from hydroxycinnamic acid, and the A ring is formed through the aceto-malonate pathway [15]. Through the stepwise condensation of three acetate units from malonyl-CoA and p-coumaroyl-CoA naringenin chalcone is formed. This reaction is catalyzed by chalone synthase (CHS) [15,25,27]. Chalcone-flavanone isomerase (CHI) catalyzes the conversion of naringenin chalcone to flavanone naringenin, which is the precursor for other flavonoid subclasses [15,25].

#### 1.2.2. Biological Activity of Phenolic Compounds

Phenolic compounds are known for many biological activities and health-promoting properties. One of the most important is their antioxidant activity. They can interrupt the radical chain reaction, since they are able to scavenge reactive oxygen species (ROS), including superoxide anion radicals, hydroxyl radicals, hydrogen peroxide, singlet oxygen and nitric oxide [15,22,28]. Similarly, phenolic compounds can chelate metal ions and form stable complexes. Therefore, they can prevent the formation of free radicals catalyzed by metal ions [13,14,21]. In addition, phenolic compounds can inhibit various enzymes (xanthine-oxidase, cyclooxygenase, nitric oxide synthase), involved in the production of free radicals [15,22,29]. The antioxidant activity is based on the presence of *ortho*-dihydroxyl groups in the B ring, which donate a hydrogen radical to a free radical and neutralise it. The antioxidant activity of flavonoids in vitro is additionally facilitated by the presence of a 4-oxo functional group in the C ring and the hydroxyl groups at positions 3 and 5 of the flavonoid structure. Furthermore, the in vitro antioxidant potential can be increased by the polymerization of monomers. One example of this is condensed tannins, which are polymers that are formed from flavan-3-ol [10,29].

Radical scavenging phenolic compounds can decrease or prevent oxidative damage to DNA caused by free radicals [22]. They can also inhibit cytochrome P450, which is involved in the conversion of some procarcinogens into reactive intermediates [28,30,31]. Similarly, phenolic compounds can inhibit cyclooxygenase and lipoxygenase, enzymes involved in platelet aggregation. Therefore, they can reduce the risk of thrombosis. In addition, phenolic compounds can inhibit the oxidation of low-density lipoprotein (LDL), which plays an important role in the development of atherosclerosis [30,31]. As a result of their many beneficial properties, phenolic compounds may be used as anti-aging agents and in the management of cardiovascular diseases, diabetes, bacterial infections, atherosclerosis, neurodegenerative diseases.

#### 1.2.3. Antimicrobial Activity of Phenolic Compounds

Phenolic compounds are also agents with antimicrobial activity, which are the focus of this article. The exact mechanism of the antimicrobial action is not yet fully known and explained. However, it is most likely based on several cellular levels. Their main targets seem to be the bacterial cell wall and membrane [20]. The bacterial cell membrane is involved in various processes: osmoregulation, respiration and transport processes, biosynthesis of peptidoglycan and lipids. Thus, any changes in the membrane structure that lead to its disruption can have negative effects on the bacterial cell [32]. By binding to the lipid bilayer, phenolic compounds can cause structural changes and ruptures in the bacterial cell membrane and thus, increase its permeability and the leakage of cellular components [20,32]. Less polar phenolic compounds part in the hydrophobic part of the membrane and more polar phenolic compounds form bonds with the polar head groups of phospholipids [32]. This can lead to a disorder of the membrane lipids, a reduction in the thickness of the bilayer and a destabilization of the membrane [32,33]. Another antimicrobial mechanism of phenolic compounds is hyperacidification. The dissociation of phenolic acids increases the proton concentration in the cytoplasm and affects the membrane potential. This leads to destabilization of the cell membrane and increases its permeability [10].

For antimicrobial activity the prevention and inhibition of biofilm formation is important [10,33]. In biofilm formation microorganisms adhere irreversibly to human tissue or medical devices and grow on the surface. Once a biofilm has formed, it is very difficult to eradicate it and kill the bacteria inside, as the polysaccharide matrix enables the transport of essential nutrients, water and oxygen to the bacterial cells, growing in the biofilm [33]. The formation of a biofilm is crucial for the survival of the bacteria. Many phenolic compounds, especially flavonoids, have antibiofilm properties. Flavonoids cause the fusion of membranes, which leads to bacterial aggregation, but then they reduce nutrient uptake into the biofilm, meaning that the bacteria can not survive. Similarly, they inhibit sortases, enzymes found in Gram-positive bacteria that catalyze the attachment of surface proteins to cell wall. These surface proteins are involved in bacterial adhesion. Thus, inhibition of sortases contributes to inhibition of biofilm formation [10,20,33,34,35]. 

One of the main targets of the antimicrobial activity of phenolic compounds is the bacterial cell wall. Peptidoglycan, a highly cross-linked polysaccharide with a pentapeptide, is an essential component of the bacterial cell wall as it is involved in maintaining the shape of the bacterial cell [20]. In Gram-negative bacteria, the peptidoglycan is located in the periplasmic space between the cytoplasmic and the outer membrane. In Gram-positive bacteria there is no outer membrane and the peptidoglycan layer is thicker [10,20]. Compared to Gram-positive bacteria, Gram-negative bacteria have an outer membrane that reduces the uptake of the phenolic compounds into the bacterial cell. Due to the lack of this additional barrier, Gram-positive bacteria appear to be more susceptible to the antimicrobial activity of phenolic compounds [20]. Phenolic compounds can interact with peptidoglycans in the membrane structure and therefore damage the bacterial cell wall, cause rupture of the cell wall and cell deformation [20]. Similarly, they can bind to certain surface proteins and enzymes that are crucial for cell wall formation. By binding to their active sites, they cause their aggregation and inactivation and therefore stop the synthesis of the cell wall [20]. For example, by binding to the active sites of D-alanine-D-alanine ligase, an enzyme responsible for the production of the peptidoglycan precursor, phenolic compounds stop the biosynthesis of cell wall [32].

Hydroxyl groups of phenolic compounds can form hydrogen bonds with the bases of nucleic acid [10]. Similarly, the hydroxyl groups and the aromatic rings of phenolic compounds interact with the amino or carboxyl groups of enzymes involved in DNA synthesis and regulation. By binding to their active sites, they cause the inhibition of topoisomerase IV or DNA gyrase and thus stop the synthesis of bacterial DNA [10,36]. They bind to the B subunit of DNA gyrase and therefore lead to a blockade of the ATP binding pocket [32]. Phenolic compounds also bind to active sites of dihydrofolate reductase, an enzyme involved in the biosynthesis of folic acid, which is a precursor of pyrimidines and purines [32]. 

Membrane potential represents an important energy source and is crucial for survival and cell growth of the bacteria. In the electron transfer chain, complexes I, II, III and IV found in the cytoplasmic membrane transfer electrons from electron donors to electron acceptors, which leads to the transfer of protons from the cytoplasm of bacterial cell across the cytoplasmic membrane. This induces an electrochemical proton gradient that enables the synthesis of ATP [33]. Phenolic compounds can interfere at various stages of the electron transfer chain and ATP synthesis, leading to reduced process of oxidative phosphorylation and thus inhibiting bacterial growth [20]. Phenolic compounds can also reduce the activity of bacterial toxins that form pores in the cell membrane of eukaryotic cells, causing their disruption [33].

Since the resistance to antimicrobial agents increases, phenolic compounds with potential activity against the efflux pump are important. Some phenolic compounds (epicatechin, epigallocatechin gallate, quercetin) demonstrated inhibitory properties against β-lactamase, which plays an important role in the antibiotic resistance [12,20].

### 1.3. Subcritical Water for the Extraction of Phenol-Rich Extracts with Antimicrobial Activity

Nowadays, antimicrobial activity is becoming more and more important due to the increasing resistance of bacteria. The discovery of new plant extracts, including those obtained from already known plants but with newer methods, is one of the important fields of research in pharmacognosy.

The aim of our study was therefore to collect and analyze the currently available articles on the use of subcritical water to obtain extracts with antimicrobial activity compared to other extraction methods. We wanted to find out how different extraction parameters influence the composition of the extract and its antimicrobial properties. The results are shown in Table 1. 

## 2. Discussion

Table 1 contains the information collected on the parameters with which each extraction method was performed and their extraction yields. The composition of the extracts is presented with the total phenolic content (TPC), the total flavonoid content (TFC) and the results of the quantification of the phenolic components by chromatographic methods. The antimicrobial activity of the extracts is indicated either by the minimum inhibitory concentration (MIC) or by the width of the inhibition zone.

**Table 1 antibiotics-13-00334-t001:** An overview of plants extracted with subcritical water to obtain phenol-rich extracts with antimicrobial properties compared to other methods.

Plant and Plant Part	Extraction Method	Extraction Conditions	Extraction Efficiency/Yield (%)	TPC	TFC	Method of Identification and Quantification and Quantified Number of Phenolic Compounds	MIC Value or Width of Inhibition Zone on Tested Bacterial Strains
*Urtica dioica* L.—leaf [37]	UAE	water, sample to solvent ratio = 1:30, t = 30 min (156 W)	18.30	mg CAE/g DW: 147.46 (±18.31)	mg of CE/g DW: 5.34 (±0.09)	qualitative analysis: UHPLC-DAD-HESI-MS/MS 24 phenolic compounds quantitative analysis: HPLC-DAD µg/g of extract: 920.97 sinapic acid (50.49), p-hydroxybenzoic acid (37.38), p-coumaric acid (18.52), ferulic acid (18.36), syringic acid (16.72); rutin (578.36), quercetin (80.65), apigenin (37.05), kaempferol (26.56), apigenin glycoside (20.00), naringenin (15.08), rosmarinic acid (11.15), luteolin (10.65)	µg/mL: positive control: amracin (A) *S. aureus* ATCC 25923: 78.12 (A: 0.97) *E. coli* ATCC 25922: 156.25 (A: 0.49) *P. vulgaris* ATCC 13315: 156.25 (A: 0.49) *K. pneumoniae* ATCC 13883: 156.40 (A: 0.97) *B. subtilis* ATCC 6633: 156.40 (A: 0.24) *P. mirabilis* ATCC 14153: 312.50 (A: 0.49)
	MAE	water, sample to solvent ratio = 1:30, t = 30 min (450 W)	30.75	380.08 (±14.91)	10.99 (±0.12)	qualitative analysis: UHPLC-DAD-HESI-MS/MS 27 phenolic compounds quantitative analysis: HPLC-DAD µg/g of extract: 1146.13 sinapic acid (63.12), p-hydroxybenzoic acid (48.68), p-coumaric acid (23.12), ferulic acid (21.07), syringic acid (20.29); rutin (722.83), quercetin (95.71), apigenin (46.34), kaempferol (32.58), apigenin glycoside (26.44), naringenin (21.46), rosmarinic acid (13.27), luteolin (11.22)	*S. aureus*: 39.10, *P. mirabilis*: 78.12, *E. coli*: 156.25, *P. vulgaris*: 312.50, *K. pneumoniae*: 312.50, *B. subtilis*: 625.00
	SWE	T = 125 °C, p = 35 bar, t = 30 min, sample to solvent ratio = 1:30	42.81	463.59 (± 15.60)	11.00 (±0.03)	qualitative analysis: UHPLC-DAD-HESI-MS/MS 22 phenolic compounds quantitative analysis: HPLC-DAD µg/g of extract: 336.23 sinapic acid (18.08), p-hydroxybenzoic acid (10.72), ferulic acid (10.30), syringic acid (7.71), p-coumaric acid (4.84); rutin (215.49), quercetin (25.51), apigenin (10.02), kaempferol (9.95), apigenin glycoside (9.95), rosmarinic acid (6.03), naringenin (4.48), luteolin (3.15)	*S. aureus*: 9.76, *E. coli*: 78.12, *K. pneumoniae*: 78.25, *P. vulgaris*: 156.25, *B. subtilis*: 312.50, *P. mirabilis*: 312.50
*Allium ursinum* L.—leaf [38]	SWE	T = 180 °C, p = 1500 psi, t = 10 min		mg GAE/g DW: 4.23	mg QE/g DW: 0.73	HPLC-DAD 11 phenolic compounds µg/mL of extract: gallic acid (32.97), gallic acid derivate (9.10), gallic acid derivate (7.24), kaempferol derivate (8.96), kaempferol derivate (16.76), kaempferol derivate (9.48), kaempferol derivate (20.45), kaempferol derivate (29.95), catechin derivate (7.24), catechin derivate (6.89), catechin derivate (3.44)	mg/mL: *L. monocytogenes*: 28 *S. enteritidis*, *E. coli* 10536, *E. coli* 8739, *P. hauseri*, *E. faecalis*: 29
*Matricaria chamomilla* L.—flower [39]	Soxhlet	70% EtOH, t = 40 min	25.75	mg CAE/g DW: 141.41	mg RE/g DW: 64.32		µg/mL: positive control: amracin (A) *E. coli* ATCC 25922: 39.10 (A: 0.97) *P. vulgaris* ATCC 13315: 78.125 (A: 0.49) *P. mirabilis* ATCC 14153: 78.125 (A: 0.49) *B. subtilis* ATCC 6633: 78.125 (A: 0.24) *S. aureus* ATCC 25923: 156.25 (A:0.97) *K. pneumoniae* ATCC 13883: 156.25 (A: 0.49)
	MAE	70% EtOH, t = 40 min	31.64	117.31	58.23		*E. coli*: 78.125, *P. mirabilis*: 78.125, *B. subtilis*: 156.25, *S. aureus*: 156.25, *K. pneumoniae*: 312.5, *P. vulgaris*: 312.5
	UAE	70% EtOH, t = 40 min	11.09	123.40	51.60		*E. coli*: 39.10, *P. mirabilis*: 39.10, *B. subtilis*: 78.125, *S. aureus*: 78.125, *P. vulgaris*: 78.125, *K. pneumoniae*: 156.25
	SWE	T = 200 °C, p = 1,6 bar, t = 40 min, sample to solvent ratio = 1:50	42.10	151.45	49.70	UHPLC-DAD-HESI-MS/MS 28 detected compounds apigenin-7-O-glucoside, caffeic acid phenylethyl ester, catechin, gallic acid, proanthocyanidin dimer, ferulic acid glucoside, 5-O-feruloylquinic acid, chlorogenic acid, luteolin, epicatechin, luteolin-7-O-glucoside, caffeoyl-hexoside-methylglutarate, quercetin-3-glucoside, 3-p-coumaroylquinic acid, pelargonidin-succinylarabinoside or pelargonidin-malonylrhamnoside, p-coumaroyl-hexosidemethylglutarate, apigenin, dicaffeolyquinic acid, pinobanksin-3-O-butyrate, hyperfirin, apigenin O-glucuronide, adhyperfirin	*E. coli*: 39.10, *P. mirabilis*: 78.125, *P. vulgaris*: 78.125, *B. subtilis*: 156.25, *K. pneumoniae*: 156.25, *S. aureus*: 312.50
*Satureja hortensis* L.—herb [40]	Soxhlet	96% EtOH, t = 8 h		mg GAE/g DW: 119.28 (±0.50)	mg RE/g DW: 5.23 (±0.76)	HPLC-DAD 15 phenolic compounds µg/g of extract: 752.54 rosmarinic acid (301.66), quercetin (155.15), apigenin (52.78), kaempferol (46.63), luteolin (40.54), chlorogenic acid (36.06), rutin (33.54), apigenin-glycoside (24.39), p-coumaric acid (14.81), p-hydroxybenzoic acid (12.87), vanillic acid (11.03), naringenin (7.51), sinapic acid (7.09), caffeic acid (6.27), ferulic acid (2.15)	µg/mL: positive control: amracin (A) *E. coli* ATCC 25922: 7.81 (A: 0.49) *S. aureus* ATCC 25923: 7.81 (A: 0.97) *L. ivanovii* ATCC 19119: 31.25 (A: 0.49) *S. typhimurium* ATCC 14028: 31.25 (A: 0.24) *E. faccalis* ATCC 2912: 62.50 (A: 0.49) *E. aerogenus* ATCC 13048: 62.50 (A: 0.49) *S. saprophiticus* ATCC 15035: 125.00 (A: 0.24) *L. inocun* ATCC 33090: 125.00 (A: 0.97) *L. monocytogenes* ATCC 19112: 125.00 (A: 0.49) *B. spieizeneii* ATCC 6633: 125.00 (A: 0.24) *S. enteritidas* ATCC 13076: 125.00 (A: 0.97) *C. freundi* ATCC 43864: 125.00 (A: 0.49) *P. aeroginosa* ATCC 27853: 125.00 (A: 0.97) *E. faccium* ATCC 6057: 250.00 (A: 0.97) *P. mirabilis* ATCC 35659: 250.00 (A: 0.49)
	maceration	96% EtOH, t = 7 days, T = 22 °C		125.34 (±0.13)	16.27 (±0.34)	HPLC-DAD 13 phenolic compounds µg/g of extract: 351.92 rosmarinic acid (287.59), chlorogenic acid (17.30), kaempferol (11.75), rutin (10.75), vanillic acid (9.02), apigenin (3.15), p-coumaric acid (2.84), p-hydroxybenzoic acid (2.35), apigenin-glycoside (2.16), quercetin (1.77), sinapic acid (1.47), luteolin (1.26), naringenin (0.46)	*E. aerogenus*: 7.81, *S. saprophiticus*: 15.82, *E. faccium*: 15.82, *S. typhimurium*: 15.82, *P. mirabilis*: 31.25, *E. coli*: 62.50, *L. inocun*: 62.50, *L. monocytogenes*: 62.50, *C. freundi*: 62.50, *S. aureus*: 125.00, *B. spieizeneii:* 125.00, *P. aeroginosa*: 125.00, *L. ivanovii*: 250.00, *E. faccalis*: 250.00, *S. enteritidas*: 250.00
	UAE	96% EtOH, t = 30 min (216 W), frequency = 40 kHz		132.40 (±0.65)	19.68 (±0.50)	HPLC-DAD 12 phenolic compounds µg/g of extract: 43.22 rutin (24.04), quercetin (6.42), sinapic acid (4.24), apigenin (1.44), rosmarinic acid (1.34), kaempferol (1.24), p-coumaric acid (1.08), ferulic acid (0.90), luteolin (0.82), apigenin-glycoside (0.82), syringic acid (0.56), naringenin (0.32)	*S. saprophiticus*: 15.82, *S. typhimurium*: 15.82, *C. freundi*: 15.82, *P. aeroginosa*: 15.82, *E. faccalis*: 15.82, *E. faccium*: 31.25, *L. inocun*: 31.25, *L. monocytogenes*: 31.25, *S. aureus*: 31.25, *B. spieizeneii:* 31.25, *L. ivanovii*: 31.25, *E. aerogenus*: 62.50, *P. mirabilis*: 62.50, *E. coli*: 62.50, *S. enteritidas*: 500.00
	MAE	t = 30 min (600 W)		147.21 (±0.15)	23.10 (±0.18)	HPLC-DAD 12 phenolic compounds µg/g of extract: 95.22 quercetin (41.26), rutin (28.48), rosmarinic acid (9.62), sinapic acid (4.88), apigenin-glycoside (2.62), apigenin (2.36), kaempferol (1.96), luteolin (1.10), ferulic acid (1.08), naringenin (0.94), syringic acid (0.48), p-coumaric acid (0.44)	*P. aeroginosa*: 31.25, *L. monocytogenes*: 31.25, *S. aureus*: 31.25, *S. saprophiticus*: 62.50, *E. faccalis*: 62.50, *S. typhimurium*: 125.00, *E. faccium*: 125.00, *L. inocun*: 125.00, *E. coli*: 125.00, *C. freundi*: 250.00, *B. spieizeneii:* 250.00, *L. ivanovii*: 250.00, *P. mirabilis*: 250.00, *S. enteritidas*: 250.00, *E. aerogenus*: 500.00
	SWE	T = 140 °C, p = 40 bar, t = 30 min		151.54 (±0.85)	28.42 (±0.29)	HPLC-DAD 12 phenolic compounds µg/g of extract: 43.32 rutin (16.56), quercetin (11.12), p-hydrohybenzoic acid (7.58), rosmarinic acid (2.66), sinapic acid (1.42), kaempferol (1.12), apigenin-glycoside (0.88), apigenin (0.78), luteolin (0.46), ferulic acid (0.26), naringenin (0.26), luteolin-glycoside (0.22)	*S. saprophiticus*: 7.81, *B. spieizeneii:* 31.25, *P. aeroginosa*: 62.50, *L. monocytogenes*: 62.50, *S. aureus*: 62.50, *S. enteritidas*: 62.50, *E. faccium*: 125.00, *L. inocun*: 125.00, *C. freundi*: 125.00, *L. ivanovii*: 125.00, *E. faccalis*: 250.00, *S. typhimurium*: 250.00, *E. coli*: 250.00, *E. aerogenus*: 250.00, *P. mirabilis*: 500.00
*Satureja kitaibelii* Wierzb. Ex Heuff—flower [41]	SWE	T = 130 °C, t = 30 min, sample to solvent ratio = 1:30	25.5			HPLC 10 phenolic compounds mg/g of extract: 89.3 syringic acid (37.88), caffeic acid (18.06), epicatechin (10.04), protocatechuic acid (7.87), vanillic acid (6.20), ferulic acid (5.93), rutin (1.68), apigenin (1.05), chlorogenic acid (0.56), luteolin (0.06)	mg/mL: positive control: cefotaxime + clavulanic acid *E. faecalis* ATCC 19433: 1.04, *S. aureus* ATCC 25923: 2.08, *L. monocytogenes* ATCC 35152: 2.08, *B. cereus* ATCC 11778: 8.325, *E. coli* ATCC 25922: > 33.3, *P. aeruginosa* ATCC 27853: > 33.3, *S. typhimurium* ATCC 13311: > 33.3
*Crocus sativus* L.—corm [42]	SWE	T: 100, 140, 180 °C, t: 10, 20, 30 min *Optimal conditions:* T = 180 °C, t = 22 min	43.55	mg GAE/g DW: 8.08	mg QE/g DW: 0.12	GC-MS	mg/mL: positive control: gentamicin *S. aureus* PTCC 1764: 150, *E. coli* PTCC 1330: 300
*Castanea sativa* Mill.—shell [43]	SWE	T: 110, 120, 140, 160, 180 °C, p = 20 bar, t = 30 min, sample to solvent ratio = 1:30	110 °C: 20.88 (±0.78) 120 °C: 21.00 (±1.49) 140 °C: 20.83 (±0.62) 160 °C: 20.97 (±1.74) 180 °C: 20.29 (±0.93)	mg GAE/g DW: 110 °C: 239.53 (±23.17) 120 °C: 201.75 (±13.02) 140 °C: 162.52(±7.14) 160 °C: 122.31 (±5.89) 180 °C: 126.19 (±6.33)		HPLC-PDA 24 detected phenolic compounds mg/g of extract: 110 °C: 19.01 120 °C: 15.58 140 °C: 11.84 160 °C: 6.75 180 °C: 6.92 3,5-dicaffeolquinic acid, 4-O-caffeyolquinic acid, caffeoylquinic acid, caftaric acid, caffeic acid, chlorogenic acid, p-coumaric acid, ellagic acid, gallic acid, ferulic acid, neochlorogenic acid, protocatechuic acid, sinapic acid, syringic acid, vanillic acid, quercetin-3-O-galactoside, quercetin-3-O-glucopyranoside, naringin, rutin, catechin, epicatechin, phloridzin, resveratrol, trans-polydatin	mg/mL: *S. aureus* MRSA: 110 °C: 4, 120 °C: 8, 140 °C: 32, 160 °C: 4, 180 °C: 4 *S. aureus* ATCC25913: 110 °C: /, 120 °C: 64, 140 °C: 4, 160 °C: 4, 180 °C: 4 *E. coli ATCC25922*: 110 °C: 8, 120 °C: 8, 140 °C: 4, 160 °C: 4, 180 °C: 8 *E. coli CTXM2*: 110 °C: 4, 120 °C: 8, 140 °C: 4, 160 °C: 4, 180 °C: / *E. coli ATCC8739*: 110 °C: 4, 120 °C: 4, 140 °C: 16, 160 °C: 4, 180 °C: 8 *E. faecalis*: 110 °C: 64, 120 °C: 2, 140 °C: 64, 160 °C: 8, 180 C: 16
*Morus nigra* L.—leaf [44]	SWE	T: 60, 100, 130, 160, 200 °C, p = 10 bar, t = 30 min, sample to water ratio = 1:40		mg GAE/g DW (160 °C): 61.89 TPC (160 °C) > TPC (200 °C) > TPC (130 °C) > TPC (100 °C) > TPC (60 °C)		HPLC-DAD 9 phenolic compounds mg/g of extract: chlorogenic acid (183), caffeic acid (91.1), protocatechuic acid (79.5), β-resorcylic acid (62.0), naringin (56.8), rutin (43.7), catechin (38.0), gallic acid (26.3), p-coumaric acid (9.57)	µg/mL: positive control: amracin (A) *S. aureus* ATCC 25923: 39.1 (A: 0.97) *K. pneumoniae* ATCC 13883: 78.125 (A: 0.49) *P. vulgaris* ATCC 13315: 78.125 (A: 0.49) *E. coli* ATCC 25922: 312.5 (A: 0.97) *P. mirabilis* ATCC 14153: 312.5 (A: 0.49) *B. subtilis* ATCC 6633: 312.5 (A: 0.24)
*Teucrium chamaedrys* L.—flower [44]	SWE	T: 60, 100, 130, 160, 200 °C, p = 10 bar, t = 30 min, sample to water ratio = 1:40		mg GAE/g DW (160 °C): 176.74 TPC (160 °C) > TPC (200 °C) > TPC (130 °C) > TPC (100 °C) ≈ TPC (60 °C)		HPLC-DAD 9 phenolic compounds mg/g of extract: gallic acid (217), catechin (101), chlorogenic acid (76.3), protocatechuic acid (73.7), caffeic acid (54.2), vanillic acid (42.2), epicatechin (38.1), ferulic acid (29.1), sinapic acid (19.2)	µg/mL: positive control: amracin (A) *S. aureus* ATCC 25923: 78.125 (A: 0.97) *K. pneumoniae* ATCC 13883: 78.125 (A: 0.49) *P. mirabilis* ATCC 14153: 78.125 (A: 0.49) *B. subtilis* ATCC 6633: 156.25 (A: 0.24) *P. vulgaris* ATCC 13315: 156.4 (A: 0.49) *E. coli* ATCC 25922: 312.5 (A: 0.97)
*Geranium macrorrhizum* L.—leaf [44]	SWE	T: 60, 100, 130, 160, 200 °C, p = 10 bar, t = 30 min, sample to water ratio = 1:40		TPC (130 °C) > TPC (100 °C) > TPC (160 °C) > TPC (60 °C) > TPC (200 °C)		HPLC-DAD 7 phenolic compounds mg/g of extract: gallic acid (1512), protocatechuic acid (234), ferulic acid (128), chlorogenic acid (106.9), catechin (97.7), vanillic acid (14.3), p-coumaric acid (8.64)	µg/mL: positive control: amracin (A) *S. aureus* ATCC 25923: 19.53 (A: 0.97) *B. subtilis* ATCC 6633: 78.125 (A: 0.24) *K. pneumoniae* ATCC 13883: 156.4 (A: 0.49) *P. vulgaris* ATCC 13315: 312.5 (A: 0.49) *E. coli* ATCC 25922: 312.5 (A: 0.97) *P. mirabilis* ATCC 14153: 312.5 (A: 0.49)
*Symphytum officinale* L.—leaf [44]	SWE	T: 60, 100, 130, 160, 200 °C, p = 10 bar, t = 30 min, sample to water ratio = 1:40		TPC (130 °C) > TPC (160 °C) > TPC (100 °C) > TPC (60 °C) > TPC (200 °C)		HPLC-DAD 11 phenolic compounds mg/g of extract: p-coumaric acid (157), protocatechuic acid (135), gallic acid (122), caffeic acid (89.6), rutin (53.5), β-resorcylic acid (30.6), naringin (28.4), ferulic acid (14.1), sinapic acid (10.6), naringenin (3.36), cinnamic acid (1.27)	µg/mL: positive control: amracin (A) *S. aureus* ATCC 25923: 39.1 (A: 0.97) *B. subtilis* ATCC 6633: 78.125 (A: 0.24) *E. coli* ATCC 25922: 156.25 (A: 0.97) *K. pneumoniae* ATCC 13883: 156.4 (A: 0.49) *P. vulgaris* ATCC 13315: 312.5 (A: 0.49) *P. mirabilis* ATCC 14153: 625 (A: 0.49)
*Aronia melanocarpa* (Michx.) Elliott [45]	SWE—leaves	T = 130 °C, t = 20 min, p = 35 bar, sample to solvent ratio = 1:20		mg CAE/g DW: 131.53 (±0.96)	mg RE/g DW: 88.64 (±0.31)	HPLC-DAD 14 phenolic compounds mg/g of extract: rutin (0.693), sinapic acid (0.547), luteolin (0.334), apigenin (0.210), rosmarinic acid (0.155), p-hydroxybenzoic acid (0.128), quercetin (0.105), p-coumaric acid (0.093), kaempferol (0.069), syringic acid (0.055), ferulic acid (0.046), vanillic acid (0.041), chlorogenic acid (0.029), naringenin (0.026)	µg/mL: positive control: amracin (A) *S. aureus* ATCC 25923: 39.10 (A: 19.53) *B. subtilis* ATCC 6633: 78.12 (A: 78.12) *E. coli* ATCC 25922: 312.50 (A: 19.53) *K. pneumoniae* ATCC 13883: 312.50 (A: 39.10) *P. vulgaris* ATCC 13315: 39.10 (A: 156.25) *P. mirabilis* ATCC 14153: 19.53 (A: 312.50)
	SWE—berries			13.88 (±0.02)	10.00 (±0.25)	HPLC-DAD 14 phenolic compounds mg/g of extract: rutin (5.544), quercetin (1.396), sinapic acid (1.072), p-hydroxybenzoic acid (0.469), syringic acid (0.419), kaempferol (0.271), apigenin (0.242), vanillic acid (0.238), rosmarinic acid (0.236), p-coumaric acid (0.175), ferulic acid (0.173), narigenin (0.172), chlorogenic acid (0.171), luteolin (0.154)	*S. aureus*: 78.20, *B. subtilis*: 312.50, *E. coli*: 312.50, *K. pneumoniae*: 312.50, *P. vulgaris*: 78.20, *P. mirabilis*: 78.125
	SWE—stems			49.96 (±0.15)	25.10 (±0.38)	HPLC-DAD 15 phenolic compounds mg/g of extract: rutin (1.264), luteolin (0.364), sinapic acid (0.290), quercetin (0.247), p-hydroxybenzoic acid (0.161), apigenin (0.151), protocatechuic acid (0.150), rosmarinic acid (0.115), naringenin (0.065), syringic acid (0.056), ferulic acid (0.051), chlorogenic acid (0.050), kaempferol (0.049), p-coumaric acid (0.033), caffeic acid (0.019)	*S. aureus*: 156.25, *B. subtilis*: 625.00, *E. cColi*: 19.53, *K. pneumoniae*: 625.00, *P. vulgaris*: 78.12, *P. mirabilis*: 312.50
*Pseuderanthemum palatiferum* (Nees) Radlk.—leaf [46]	MeOH extraction	T = 25 °C, t = 19 h	g/g: 0.06 (±0.01)	mg CE/g DW: 6.94 (±0.54)	mg RE/g DW: 6.22 (±0.16)	HPLC mg/g of extract: apigenin: 1.90 kaempferol: 0.73	Inhibition zone (mm): positive control: ampicillin >6 mm: *S. aureus* ATCC 6538P, *E. coli* ATCC 25922, *P. putida* 10464, *P. aeruginosa* KCCM 9027, *L. monocytogenes* ATCC 7644
	hot water extraction	T = 80 °C, t = 30 min	0.34 (±0.02)	15.27 (±0.11)	14.71 (±0.14)	apigenin: 2.05 kaempferol: 0.89	>6 mm: *S. aureus*, *E. coli*, *P. putida*, *P. aeruginosa*, *L. monocytogenes*
	Soxhlet	70% EtOH, t = 7 h	0.13 (±0.04)	10.77 (±0.72)	8.28 (±0.26)	apigenin: 2.11 kaempferol: 0.63	>6 mm: *S. aureus*, *E. coli*, *P. putida*, *P. aeruginosa*, *L. monocytogenes*
	SWE	T: 130, 150, 170, 190, 210, 230, 250, 270 °C, p = 80 bar, t = 15 min, solid liquid ratio = 1:70	110 °C: 0.31 (±0.02) 130 °C: 0.32 (±0.01) 150 °C: 0.39 (±0.01) 170 °C: 0.43 (±0.02) 190 °C: 0.44 (±0.01) 210 °C: 0.46 (±0.01) 230 °C: 0.48 (±0.01) 250 °C: 0.49 (±0.01) 270 °C: 0.50 (±0.01)	110 °C: 23.03 (±0.79) 130 °C: 24.35 (±0.45) 150 °C: 28.98 (±1.01) 170 °C: 29.47 (±0.04) 190 °C: 33.68 (±0.29) 210 °C: 25.29 (±0.02) 230 °C: 23.09 (±0.43) 250 °C: 18.53 (±1.76) 270 °C: 9.48 (±2.38)	110 °C: 18.94 (±0.80) 130 °C: 20.71 (±0.42) 150 °C: 19.38 (±0.63) 170 °C: 18.38 (±0.15) 190 °C: 18.33 (±0.43) 210 °C: 16.58 (±0.12) 230 °C: 15.27 (±0.36) 250 °C: 11.47 (±0.41) 270 °C: 8.16 (±0.13)	110 °C: apigenin: 2.63; kaempferol: 1.44 130 °C: apigenin: 2.93; kaempferol: 1.61 150 °C: apigenin: 3.23; kaempferol: 1.90 170 °C: apigenin: 3.46; kaempferol: 2.31 190 °C: apigenin: 3.34; kaempferol: 2.43 210 °C: apigenin: 2.80; kaempferol: 1.92 230 °C: apigenin: 2.47; kaempferol: 1.53 250 °C: apigenin: 1.14; kaempferol: 1.25 270 °C: apigenin: 0.97; kaempferol: 1.01	110 °C: >6 mm: *S. aureus*, *E. coli*, *P. putida*, *P. aeruginosa*, *L. monocytogenes* 130 °C: >6 mm: *S. aureus, E. coli, P. putida, P. aeruginosa, L. monocytogenes* 150 °C: >6 mm: *P. putida*, *L. monocytogenes;* >8 mm: *S. aureus*, *E. coli*, *P. aeruginosa* 170 °C: >8 mm: *S. aureus*, *E. coli*, *P. putida*, *P. aeruginosa*, *L. monocytogenes* 190 °C: >8 mm: *S. aureus*, *E. coli*, *P. putida*, *P. aeruginosa*, *L. monocytogenes* 210 °C: >8 mm: *S. aureus*, *E. coli*, *P. putida*, *P. aeruginosa*, *L. monocytogenes* 230 °C: >8 mm: *S. aureus*, *E. coli*, *P. putida*, *P. aeruginosa*, *L. monocytogenes* 250 °C: >6 mm: *S. aureus*, *E. coli*, *P. putida*, *P. aeruginosa*, *L. monocytogenes* 270 °C: >6 mm: *S. aureus*, *E. coli*, *P. putida*, *P. aeruginosa*, *L. monocytogenes*

**Notes:** CAE—chlorogenic acid equivalents, CE—catechin equivalents, GAE—gallic acid equivalents, QE—quercetin equivalents, RE—rutin equivalents, DW—dry weight.

### 2.1. Influence of Extraction Methods on Phenol and Flavonoid Content

In most of the reviewed studies, the content of phenolic compounds was analyzed through spectrophotometric determination of TPC and TFC content, as well as qualitative and quantitative chromatographic analysis using HPLC and/or GC-MS. 

In most of the studies, TPC values determined by the Folin-Ciocalteu (FC) assay were higher after extraction with subcritical water than after UAE or MAE extraction or conventional methods (Soxhlet extraction, maceration, percolation, hot water extraction), regardless of the solvent used (MeOH, EtOH). 

The most significant difference was observed in the study of *U. dioica* leaves, where the TPC value of the SWE extract was 3.1-fold higher than that of the UAE extract; the extraction solvent was water in both extractions [37]. Slightly less pronounced differences were obtained in the extraction of *M. chamomilla* flowers and *S. hortensis* herb, where comparative extractions were performed with 70% EtOH and 96% EtOH, respectively [39,40]. A study on *P. palatiferum* leaves by Ho&Chun does not allow a detailed comparison, as the comparative extractions were performed with different solvents (70% EtOH, MeOH), temperatures and times [46]. However, it is plausible that SWE is the method of choice for the extraction of phenolic compounds in terms of the TPC parameter under optimal extraction conditions. The TPC results correlated with the TFC results in the studies of *U. dioica*, *S. hortensis* and *P. palatiferum*, but not of *M. chamomilla*.

The most important parameter when using subcritical water as an extraction solvent is the temperature at which the extraction is performed. Usually, the final yield of the extraction increases with increasing temperature [39,46]. The extraction of *P. palatiferum* leaves was carried out at nine different temperatures between 110 and 270 °C. The extraction yield increased with increasing temperature (from 0.3 to 0.5%); but the TPC and TFC values did not increase accordingly. At 110 °C, the TPC (expressed as mg CE/g DW) and TFC (expressed as mg RE/g DW) values were 23.03 and 18.94, respectively. The latter increased in the temperature range up to 190 °C (33.68 and 18.33, respectively), then gradually decreased to the values obtained at the initial extraction temperature, but dropped to a minimum of 9.48 and 8.16, respectively, at a temperature of 270 °C [46]. At higher temperatures the penetration of the subcritical water is generally easier, and the dissolution of the compounds is faster, resulting in higher TPC and TFC values. However, at very high temperatures, the TPC and TFC values start to decrease, which is most likely due to thermal degradation of the phenolic compounds [42,44,46]. Similarly, the degradation of unstable phenolic compounds at higher temperatures could also be the reason why the TPC values determined in *C. sativa* shell extract decreased with increasing temperatures; at 110 °C the TPC value was 239.53 mg GAE/g DW and at 180 °C it was 126.2 mg GAE/g DW [43].

In some cases, however, the results of the two spectrophotometric tests differed significantly from the results obtained by identification and quantification of the phenolic compounds by HPLC. For example, in the extraction of *U. dioica* leaves, the quantified number of phenolic compounds in the final extract was highest after MAE (1146.1 μg/g extract) and lowest after SWE (336.2 μg/g extract); the TPC and TFC values were highest after SWE [40].

Similarly, in the extracts obtained from *S. hortensis* herb, the highest TPC and TFC values were found after SWE, and the lowest after Soxhlet extraction with ethanol as the extraction solvent. However, using HPLC, the highest content of phenolic compounds was determined in the extract obtained with Soxhlet extraction (753 µg/g extract), while it was only 43.3 µg/g extract in the extract obtained with SWE [40]. 

It is important to note that the determination of TFC with the FC assay is limited, as the FC reagent is not only specific for the determination of phenolic compounds, but also reacts with many other compounds in the sample such as amino acids, nucleotides, unsaturated fatty acids, proteins and vitamins, which can lead to higher final TFC values [37]. The reduction in surface tension and viscosity of water at a higher SWE temperature allows easier penetration into the interior of the plant material and faster dissolution of the compounds. In addition, as the value of the dielectric constant decreases, so does the polarity, which is why water under subcritical extraction conditions is also a good solvent for less polar compounds that interfere with the FC reagent and contribute to a higher final TPC value [37,44,47]. The determination of TFC using colorimetric methods is also limited because of low specificity of the reagents used to form complexes with flavonoids. The TFC test with AlCl_3_ as reagent is based on the measurement of absorbance at a wavelength of about 420 nm. However, not all flavonoids form complexes at this wavelength. In addition, flavonoids in plant samples are often present in the form of glycosides, which influence the formation of complexes with the reagent [48].

Conversely, we must be aware that chromatographic methods are only relevant if they are based on a comparison of retention time and spectrum between a target compound and a corresponding reference compound. The compounds that are present in the extract but not detected are therefore not quantified but contribute to the TPC and TFC values.

### 2.2. Influence of Extraction Methods on Yield

SWE proved to be the most efficient in terms of extraction yield compared to other conventional or advanced extraction methods. For the extracts from *U. dioica* leaves and *M. chamomilla* flowers, SWE gave a yield of 42.8% and 42.1%, respectively, and was 5.2-fold and 3.8-fold more efficient than UAE which gave the lowest yield in both studies [37,39].

The components of the plant material can undergo various chemical transformations after contact with subcritical water, as water has a high hydrolytic and oxidation potential under high temperature and pressure. This can lead to the formation of new components which, in turn, can increase the final yield. It has been shown that high temperatures and high pressure cause the degradation of lignin, cellulose and hemicellulose, which can lead to the formation of new water-soluble phenolic compounds. This can explain higher extraction yields and higher TPC values determined after using subcritical water [37,39,46,49]. When discussing the yield and efficiency of the extraction, it should also be taken into account that the times of the individual extractions in the analyzed studies were very different. While SWE, UAE and MAE took about 30 min, the conventional Soxhlet extraction and maceration methods required several hours and even several days [40].

### 2.3. Influence of Extraction Methods on Antimicrobial Activity

In most cases, the results of the antimicrobial activity of extracts containing phenolic compounds were reported as the lowest concentration of an extract at which the growth of the microorganism investigated was inhibited (MIC). The lower the MIC value, the more effective the antimicrobial activity of the extract. The diffusion method was also used, where the results were given by the size of the inhibition zone. The wider this was, the higher the antimicrobial activity of the extract. To control the sensitivity of the tested bacteria, different positive (amracin, ampicillin, cefotaxime and clavulanic acid, gentamicin) and negative (DMSO, distilled water) were used.

The studies reviewed confirm that the extracts obtained with modern green technologies generally have good antimicrobial activity. SWE extracts of *U. dioica* leaves showed stronger antimicrobial activity compared to UAE and MAE extracts in three out of six bacterial cultures tested. The lowest MIC value for the extract obtained with SWE was found in the cultures of *S. aureus* (9.76 µg/mL)*, K. pneumoniae* (78.25 µg/mL) and *E. coli* (78.12 µg/mL). The antimicrobial activity is most likely due to the presence of various phenolic compounds in the extract. Indeed, TPC of the *U. dioica* extracts was highest in the SWE extract. Zeković et al. also discuss that the stronger antimicrobial activity of the SWE extracts could be the result of a higher number of compounds present in the extract acting synergistically and thus increasing the strength of the antimicrobial activity of the extract [37,44].

The antimicrobial activity of all four extracts obtained from *M. chamomilla* was tested on six different bacterial cultures: *E. coli*, *P. vulgaris*, *P. mirabilis*, *B. subtilis*, *S. aureus* and *K. pneumoniae.* As described above, the different bacterial cultures vary in sensitivity to the extracts obtained by different extraction technologies. All extracts had the strongest effect on *E. coli*. The MIC values determined for the Soxhlet, UAE and SWE extracts were 39.10 µg/mL, while the MIC value of MAE extract was 78.125 µg/mL. Similarly, the bacterial cultures of *K. pneumoniae* and *P. vulgaris* were more sensitive to Soxhlet, UAE and SWE extracts than to MAE extract. The sensitivity of *S. aureus* to UAE extract (MIC = 78,125 µg/mL) is comparable to that of *B. subtilis* to Soxhlet and UAE extract and is lower than the sensitivity of *P. mirabilis* to UAE extract (MIC = 39.10 µg/mL) [39]. 

*S. hortensis* extracts were tested on 15 different bacterial cultures. Determined MIC values show that *S. aureus* and *S. saprophiticus* are most sensitive to antimicrobial activity of tested extracts. MIC values range from 7.81 for SWE extract on *S. saprophiticus* and for Soxhlet extract on *S. aureus* to 125 for Soxhlet extract on *S. saprophiticus* and for macerate on *S. aureus*. Conversely, *S. enteritidas* is least sensitive to tested extracts, as its MIC values range from 62.50 for the SWE extract to 500 for the UAE extract [40].

*P. palatiferum* extracts obtained with SWE, Soxhlet, methanol and hot water exhibited no antimicrobial activity on *B. cereus* and *K. pneumoniae*, while antimicrobial activity was detected on other strains (*S. aureus*, *E. coli*, *P. putida*, *P. aeruginosa*, *L. monocytogenes*). SWE extracts obtained at a temperature of 110 °C had weaker antimicrobial activity than extracts obtained at a temperature of 190 °C; the width of the inhibition zones at 110 °C was wider than 6 mm, at 190 °C it was wider than 8 mm. At temperatures higher than 250 °C, the SWE extracts exhibited weaker antimicrobial activity as the width of the inhibition zones was wider than 6 mm; this is also in consistent with the decreasing TPC and TFC values due to the degradation of phenolic compounds [39,40,46]. For SWE from *C. sativa* shells, where extraction temperatures were lower, 110 to 180 °C, inconsistent fluctuation of MIC values was observed [43].

We therefore conclude that plant extracts obtained using the conventional methods reviewed, such as Soxhlet extraction, maceration and percolation—as well as extracts obtained using newer extraction technologies such as subcritical water, ultrasound or microwaves—demonstrate good antimicrobial activity. This is particularly important in view of the growing trend towards the use of newer, green extraction technologies, which are more environmentally friendly due to the lower energy consumption, the absence of large amounts of organic solvents and the lower probability of the presence of organic solvents in the final extract, as well as higher final yield [29]. It should be emphasized that the sensitivity of different bacterial strains to extracts obtained with different extraction technologies differs significantly [40].

This is also evident when comparing the strength of antimicrobial activity of extracts obtained from different plants with the same extraction solvent and under the same extraction conditions. SWEs were performed from the *M. nigra*, *S. officinale* and *G. macrorrhizum* leaves, as well as from the *T. chamaedrys* flowers. All four SWE extracts showed the strongest antimicrobial activity against *S. aureus*. When comparing the MIC values, the strongest antimicrobial activity was determined for the extract of *G. macrorrhizum*, as the MIC value for the culture of *S. aureus* was 19.53 µg/mL. HPLC was also used to determine the highest number of phenolic compounds in the extract of *G. macrorrhizum*. The extract was particularly rich in gallic acid; the quantified amount in the SWE extract was 1512 mg/100 g of dry extract. The TPC value for *G. macrorrhizum* extract was not consistent with the quantified number of phenolic compounds in the extract, as the TPC value of the *G. macrorrhizum* extract determined at the optimum temperature of 160 °C was lower than that of the extracts of *T. chamaedrys* and *S. officinale*. We conclude that in extracts obtained with the same extraction method and under the same conditions, a higher number of phenolic compounds leads to strong antimicrobial activity [44].

### 2.4. Antimicrobial Activity of Phenolic Compounds—SAR

Some authors have focused on the importance of substituted hydroxyl groups in phenolic compounds [10]. More hydroxyl groups usually increase the strength of antimicrobial activity [10,50]. One possible explanation for this is the theory of a delocalized electron system that causes the hydroxyl groups on the aromatic ring to become more acidic [51]. After entering the cell, the hydroxyl group in the cytoplasm of the bacterial cell releases a proton and is converted into a dissociated form. The increase in proton concentration in the cytoplasm therefore decreases the gradient across the membrane of the bacterial cell, which leads to its destabilization [50,51]. For the predicted mechanism of action of phenolic compounds is crucial—not only the presence of a hydroxyl group, but also a conjugated double bond system [50,52].

In *G. macrorrhizum* extract, compounds of gallic acid and protocatechuic acid, among others, have been identified [44]. Gallic acid was also quantified in the highest concentrations in SWE extracts of *A. ursinum* and *T. chamaedrys* [38,44]. Gallic acid (**1** in Figure 9) is a hydroxybenzoic acid with hydroxyl groups at *para* and both *meta* positions. The structure of protocatechuic acid (**2**) is similar to the structure of gallic acid, but without a hydroxyl group at one *meta* position. In UAE, MAE and SWE extracts of *U. dioica*, the most important phenolic compounds were p-hydroxybenzoic acid (**3**) and sinapic acid (**4**), which is a hydroxycinnamic acid with a hydroxyl group at *para* position and methoxy groups at both *meta* positions [37]. In the SWE extract of *S. kitaibelii* syringic acid and caffeic acid have been identified among other compounds [41]. Syringic acid (**5**) is a p-hydroxybenzoic acid with two additional methoxy groups at both *meta* positions, while caffeic acid (**6**) is a p-hydroxycinnamic acid with one hydroxyl group at the *meta* position. The most important phenolic compounds in the SWE extracts of *S. officinale* were p-coumaric (p-hydroxycinnamic acid) (**7**) and protocatechuic acid, while in the extracts of *M. nigra* they were caffeic and chlorogenic acid (**8**), which is a caffeic acid ester [44]. The antimicrobial activity of hydroxybenzoic and hydroxycinnamic acids mentioned above is probably based on carboxyl, hydroxyl and methoxy functional groups in their structures, which act as proton acceptors or proton donors [53]. They bind directly to the active sites of bacterial proteins through hydrogen bonds, which can lead to their inactivation [54].

Ten phenolic compounds, namely six phenolic acids and four flavonoids, were identified in the extract from the flowers of *S. kitaibelii* obtained with SWE. The flavonoid epicatechin (**9** in Figure 10) was present in the highest concentration, and rutin (**10**), luteolin (**11**) and apigenin (**12**) were also identified in the extract. The extract displayed antimicrobial activity on various bacterial cultures [41]. Among others, the flavonoids rutin, quercetin (**13**) and apigenin were identified in extracts of *U. dioica* [37]. Two flavonoids, apigenin and kaempferol (**14**), were identified in extracts of *P. palatiferum* [46]. According to Farhadi et al., at least one hydroxyl group in the A ring (especially at position 7) of a flavonoid structure is essential for the antimicrobial activity. In addition, hydroxyl groups at other positions increase activity, as they bind to the active sites of bacterial proteins [55]. All the flavonoids mentioned above have hydroxyl groups at positions 5 and 7 of the flavonoid skeleton. Similarly, the hydroxyl group at position 5 forms a hydrogen bond with the carbonyl group at position 4 of the flavonoid structure and stabilizes it [52]. Hydroxyl groups at the positions *meta* and *para* of the phenyl ring also contribute to the antimicrobial activity [55,56]. Rutin, luteolin, apigenin, naringenin (**15**), quercetin and kaempferol have a carbonyl group at position 4. With the hydroxyl group at position 5 of the flavonoid skeleton, this carbonyl group can strengthen the system of conjugated bonds, which further increases the strength of antimicrobial effect [16]. In addition, in the structures of rutin, apigenin, kaempferol and luteolin, the carbon atoms at positions 2 and 3 are linked by a double bond. Conversely, naringenin and epicatechin do not contain a double bond between carbon atoms at positions 2 and 3. According to Halake et al. hydrogenation of a double bond in the flavonoid skeleton reduces the activity of flavonoids against bacteria [56]. Together with the carbonyl group at position 4, the double bond C2-C3 presents an additional conjugated system within the structure [52].

The relationship between the chemical structure of phenolic compounds with slightly higher molecular mass—such as phenolic acids and flavonoids—and their potential antimicrobial activity is not yet fully understood, but several authors also point to the influence of increased lipophilicity [35]. Indeed, lipophilicity has been found to be a key factor of the antimicrobial activity of flavonoids against Gram-positive bacteria [57]. Lipophilicity is related to the ability of a compound to penetrate the phospholipid layer in the cell membrane of a bacterial cell. As the lipophilicity of flavonoids increases, their affinity for the lipid bilayer also increases [57,58]. A study by Yuan et al. concluded that the addition of alkyl chains to a flavonoid structure will increase their lipophilicity and thus their interactions with phospholipids in the bacterial cell membrane [57]. Interactions of flavonoids and phospholipidic bilayer can lead to changes in membrane properties. Similarly, Farhadi et al. found that the addition of prenyl groups, alkylamino chains and oxygen- or nitrogen- containg heterocylic compounds to the flavonoid skeleton generally increased its antimicrobial activity [55]. In addition, Sarbu et al. concluded that flavonoids with higher lipophilicity could interfere with the adhesion step of biofilm development [59]. A more lipophilic compound can more easily pass through the cell membrane into the interior of the bacterial cell. However, passage through the cell membrane is difficult for oversized molecules. Too many or too long alkyl chains can make it difficult for the flavonoid to penetrate the phospholipid bilayer. Conversely, too many polar hydroxyl groups in the flavonoid structure can make it difficult for them to penetrate the hydrophobic region of the cell membrane [57]. In addition, lipophilic molecules can form hydrophobic interactions with the phospholipids in the cell membrane of the bacterial cell, which contributes to its destabilization [10].

## 3. Conclusions

We can conclude that newer extraction approaches using ultrasound, microwaves or subcritical water allow extracts with antimicrobial activity to be obtained. The activity against different bacterial cultures is most likely due to the presence of phenolic compounds. The higher the content of phenolic compounds in the final extracts, the stronger their antimicrobial activity. A direct comparison of antimicrobial activity based on the extraction technology used is difficult. Nevertheless, we can confirm that extracts obtained with subcritical water have a higher extraction yield compared to conventional methods. In the future, it would be useful to analyze in more detail a possible correlation between the strength of the antimicrobial effect and the composition of the extract obtained. This way, it would be possible to define more precisely how the profile of the identified compounds in the extract and their chemical structure influence the antimicrobial activity against different bacterial cultures. It would also be useful to define the mechanism by which each identified compound with potential antimicrobial activity acts on a bacterial cell.

## Figures and Tables

**Figure 1 antibiotics-13-00334-f001:**
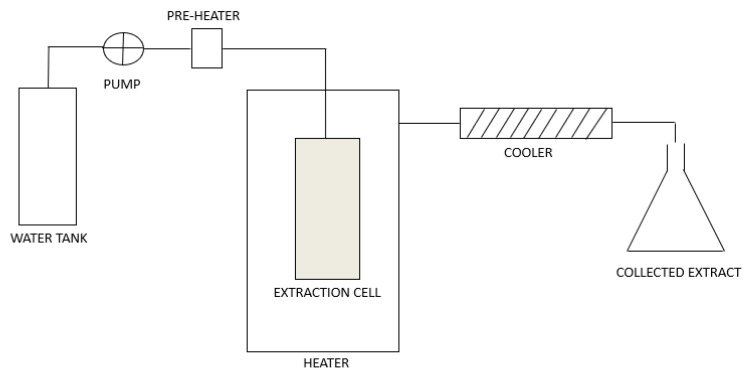
Schematic diagram of the SWE system.

**Figure 2 antibiotics-13-00334-f002:**
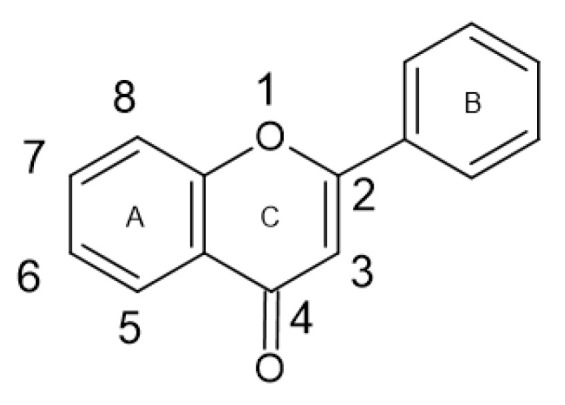
Flavone skeleton with rings A, B and C.

**Figure 3 antibiotics-13-00334-f003:**
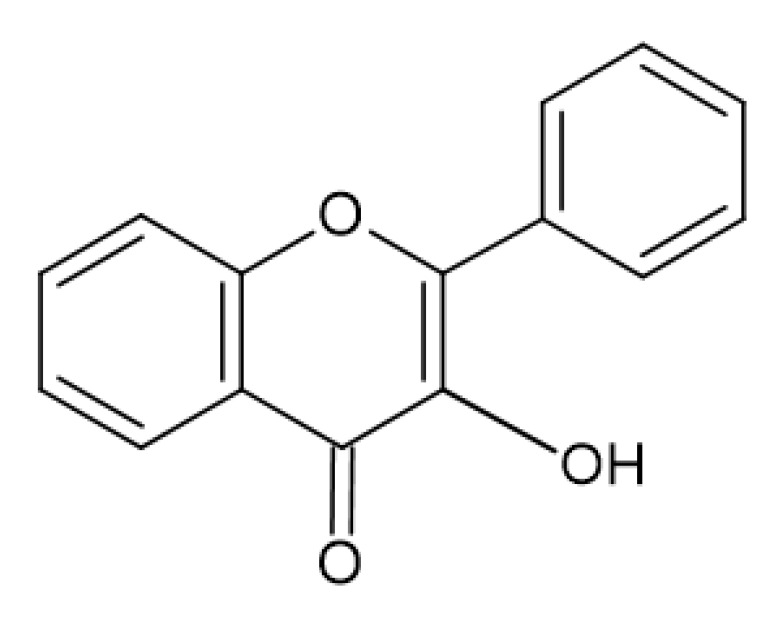
Flavonol skeleton.

**Figure 4 antibiotics-13-00334-f004:**
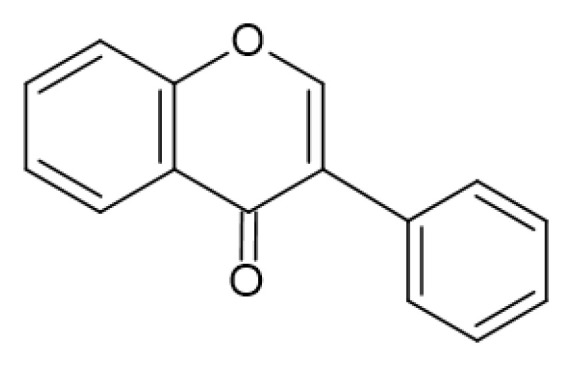
Isoflavone skeleton.

**Figure 5 antibiotics-13-00334-f005:**
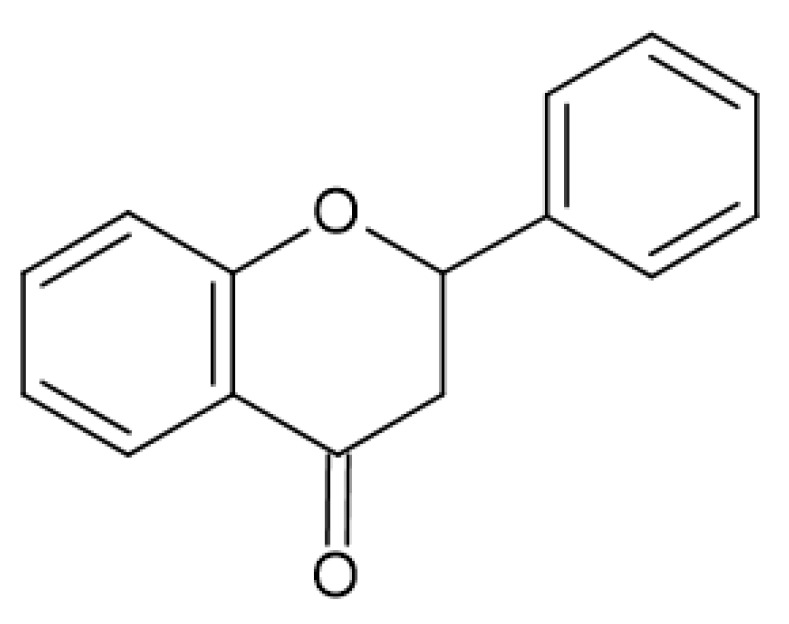
Flavanone skeleton.

**Figure 6 antibiotics-13-00334-f006:**
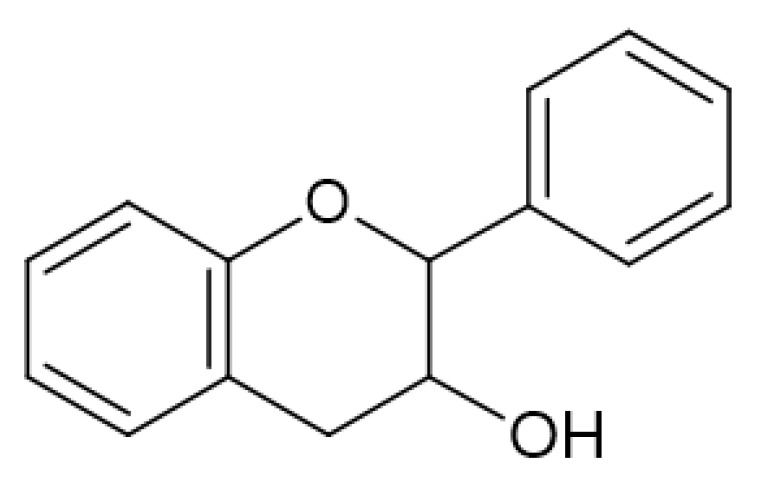
Flavanol skeleton.

**Figure 7 antibiotics-13-00334-f007:**
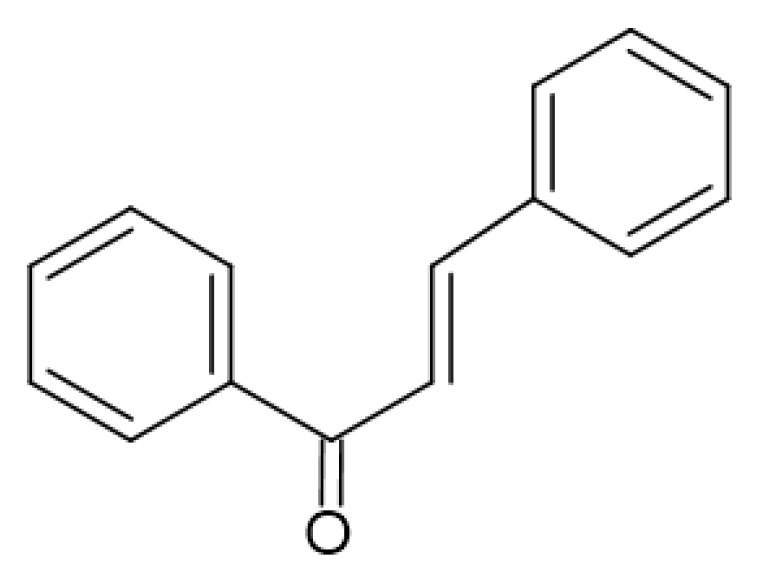
Chalcone skeleton.

**Figure 8 antibiotics-13-00334-f008:**
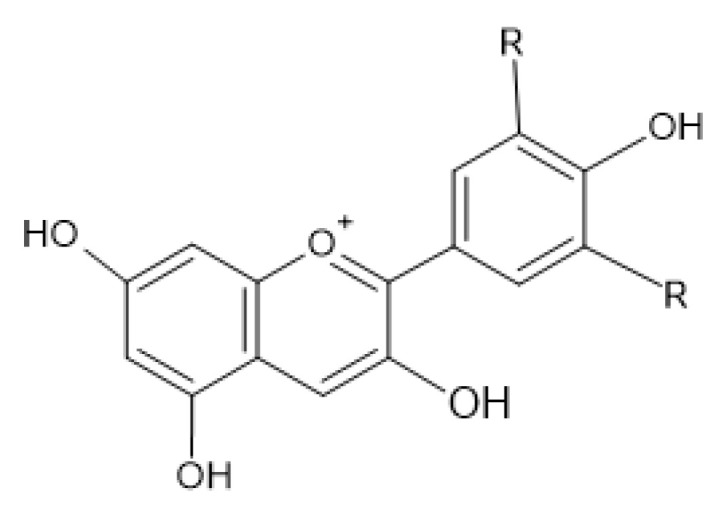
Anthocyanidin skeleton.

**Figure 9 antibiotics-13-00334-f009:**
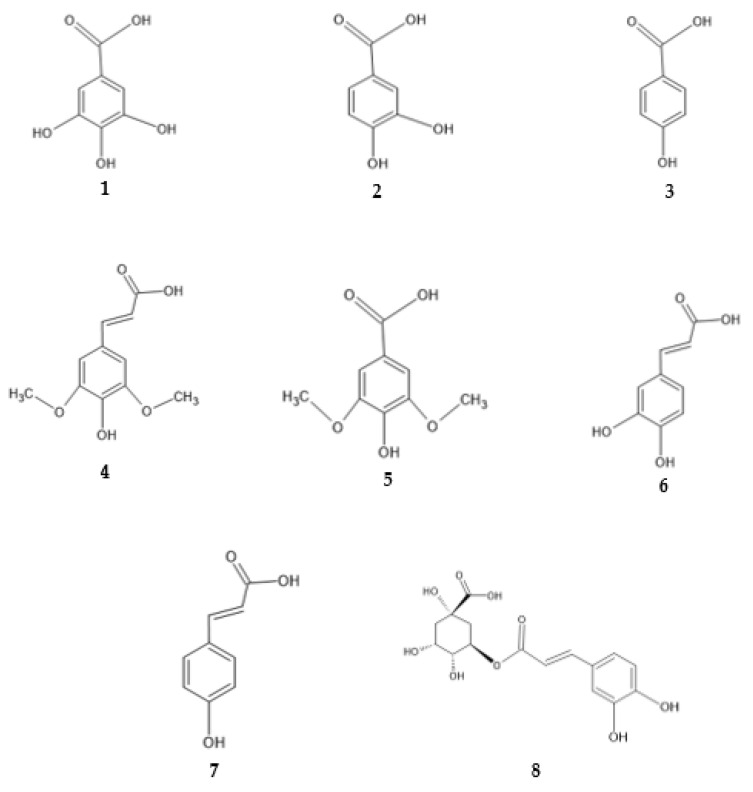
Chemical structures of **1** gallic acid, **2** protocatechuic acid, **3** p-hydroxybenzoic acid, **4** sinapic acid, **5** syringic acid, **6** caffeic acid, **7** p-coumaric acid and **8** chlorogenic acid.

**Figure 10 antibiotics-13-00334-f010:**
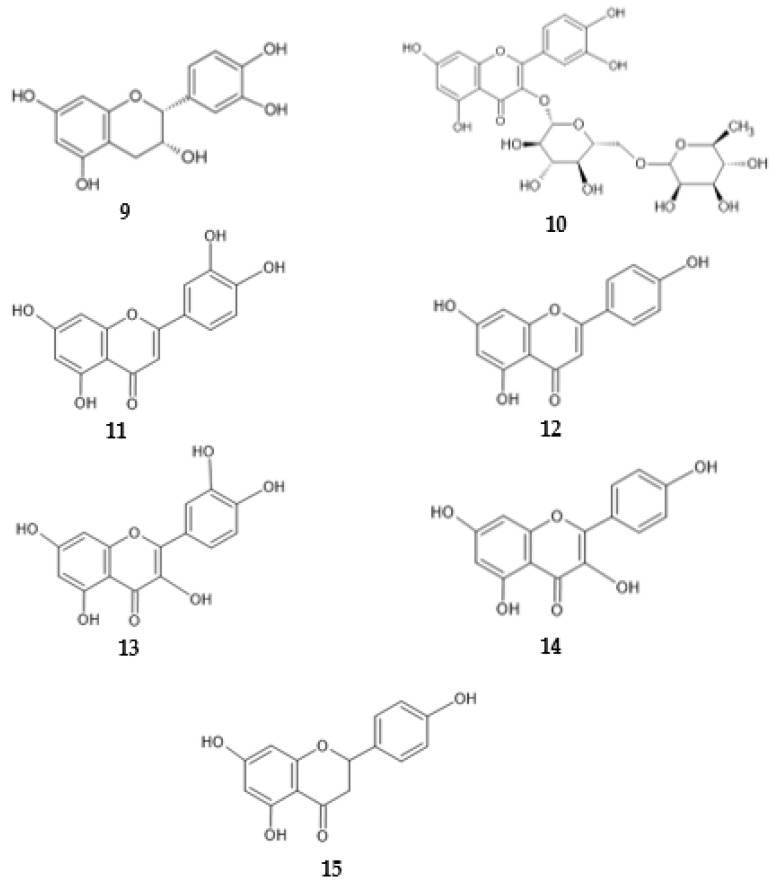
Chemical structures of **9** epicatechin, **10** rutin, **11** luteolin, **12** apigenin, **13** quercetin, **14** kaempferol and **15** naringenin.

## Data Availability

The raw data is available upon request.

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
