# Peer review of "Using Subcritical Water to Obtain Polyphenol-Rich Extracts with Antimicrobial Properties"

_antibiotics, 2024, doi:10.3390/antibiotics13040334_

Round 1

Reviewer 1 Report

Comments and Suggestions for Authors

This manuscript described a review covering a comparison between extraction by subcritical water extraction and other conventional methods,  in term of yield , content and the antimicrobial potential  of polyphenolic compounds. Therefore it may deserve publication in antibiotics.

Before a possible acceptance, this paper needed a strong revision:

The authors described the purpose of their review as the “ analyse the currently available articles on the use of subcritical water as a solvent to obtain extracts with antimicrobial activity in  relationship with findings from our experimental experiences”

A paragraph describing the methodology of reviewing is missing, precising the name of literature databases, the periode covered by the search of papers , the key words……etc The authors should explain how from the retriewed papers selected by browsing title and abstracts , they  selected 38 references . 

L122 the authors have written: The aim of our study was therefore to collect and analyse the currently available articles on the use of subcritical water to obtain extracts with antimicrobial activity………..see table 1

Since you introduced other extraction methods UAE, MAE……and conventional methods as seen in table 1,  this sentence has to be revised.

Another point is how and why did the authors select 15 plant species for their review. 

Ma be a column with the chemical composition of these plants will be informative  to distinguish better between the polar and less polar compounds. Did these plants contain essential oils?

Are there special equipment to do the SWE? A shematic  diagram of a SWE system could well illustrate the purpose of the authors. 

The title of table 1 is not appropriated   “A review of available analyses of using subcritical water to obtain extracts with phenolic compounds with potential antimicrobial properties”, because in this table the authorsmention also other methods UAE, MAE,  subcritical fluid extraction, and conventional extraction procedures. Furthermore, there are plants with saponins contents , and not phenolic compounds (hedera nepalensis)

In table 1, there are several other points which have to be improved

Could you homogeneize all the data with mean  values +/- standard deviation. For all the values and SED , 2 numbers after comma  are appreciated . you will find that in the original papers.

Furthermore in the column of MIC values, could you add when possible the name of the positive control with its values of MIC. 

For the bacteria, could you add the strain in the following manner:  capital letter-space-number

For Castanea sativa  , you include caffeine as phenolic compound !!!!! could you check the structure of caffeine,  which is a pseudoalkaloid with a  trimethyl xanthine skeleton , but absolutely not a phenolic compound. 

Probably it is rather “ caffeic acid “ which is a phenolic acid.

Usually you gave information on the quantitative analysis by HPLC/DAD of phenolic compounds expressed as microg/g of extract but for Allium ursinumwhy in microg/mL of extract?

For all plant names could you add after the Latin name the abbreviation of the taxonomist who identified the plant the first time.

For Helicteres Isora-------àHelicteres isora :

Which plant part has been used for extraction purposes?

No data are given for the phenolic and flavonoid qualitative and quantitative analysis? 

I’m afraid that this plant is not in good agreement with the title of the manuscript.

Same remark for Kunzea ericoides and Haematococcus pluvialis

In these case the antimicrobial activity is not due to phenolic compounds  and these plants are not in agreement with the titles and aims of your study

The number of identified phenolic acid and flavonoids compounds  in the cited plants is not so high ,  and therefore it will more informative to mention all of them instead of dotted like ………

Furthermore,  a  figure with the formulas of all the phenolic acids and flavonoids will be appreciated , instead of drawing    3 well known skeletons. 

L202 to 203. ---> revise because  a flavan (3,4-dihydro-2-phényl-2H-1-benzopyrane)

is not in agreement with the formula of figure 3

the   basic structure of a flavonoid is the flavone , a phenyl-2 chromone which is correct in fig 3

Discussion 

Line 34 :     The value of TFC is expressed as mg rutin Equivalent /g    not as indicated g/g

Throughout the discussion  L 34-44 the expression of the values of TPC and TFC results should included for  a better understanding,    not only numbers because the expression of results change from one experiment to another

L 257 “the antimicrobial activity is mostly due to phenolic compounds “   but in table 1  there are compounds which doesn’t contain phenolic compounds , but possess antimicrobial activity . Therefore another class of phytochemical could be responsible of the effects. This point should be better elucidated in the text.  

Author Response

Dear Reviewer,

Thank you for giving us the opportunity to submit a revised draft of our manuscript titled Using Subcritical Water to Obtain Polyphenol-Rich Extracts with Antimicrobial Properties. We appreciate the time and effort that you dedicated to providing your valuable feedback on our manuscript. We have been able to incorporate changes to reflect most of the suggestions provided by the reviewers, please find them highlighted within the manuscript. We have provided a point-by-point response to your comments and concerns. Please see the attachment.

We look forward to hearing from you, and to responding to any further questions and comments you may have.

Sincerely,

Tjaša Žagar, Rok Frlan, Nina Kočevar Glavač

Reviewer 2 Report

Comments and Suggestions for Authors

I think the topic of this review article is interesting but some areas should be improved as follows:

In Introduction the information from Lines 125-131 belongs to Results. Also, I do not think it is appropriate that Results consists of one table only, without any other text.

In Discussion the lines should be corrected.

In Line 121 (in Introduction) and Lines 58, 83 (in Discussion) the authors should cite the investigations for the relevant  paragraph. In general,  I think it is not good practise that there are not citations in whole paragraphs even if they are short and the citations are placed in the folowing paragraph.

In Lines 224-237 there is not a single citation as well.

I think the information in Lines 100 and 102 is redundant because the facts are well known.

I think a scheme (figure) of SWE (and maybe the most often used extraction methods) could be useful, so that the article can become more easily understood by the interested readers. 

This review included 38 citations only. This is very insufficient for review article. Even if SWE is not often used as extraction method and there are not enough experimental data, other areas such as the antimicrobial activity of phenolic compounds are more well described in the literature. The authors should find such areas to incorporatе more literature data that could enrich the review.

Author Response

(The authors gave the same response as above.)

Reviewer 3 Report

Comments and Suggestions for Authors

This research is important and brings valuable information for further research and with practical application.

Therefore, the paper is of interest, but some points must be considered prior acceptance:

The data presented in the introduction should highlight the novelty and originality of this research. In the introduction, the study design is not presented, with a clear presentation of the investigation steps used, of the monitored parameters.

In the introduction, more information related to antibacterial action mechanisms (including antibiofilm action), polyphenolic compounds (structure, structural classes, action mechanisms including antimicrobial) would be needed.

The terms must be written in full at the first introduction in the text and then the abbreviation used in the rest of the manuscript (eg SAR-? line 177)

Regarding Iconography. Table 1 must be improved, corrected and completed. Thus, the title of the table is not correct A review of available analyzes of using subcritical water to obtain extracts with phenolic compounds with potential antimicrobial properties.- should be added compared to other methods in the title. It would be necessary to add Notes with the abbreviations made, the units of measure for the monitored parameters, the expressions for TPC, TFC (for example mg of chlorogenic acid equivalents per gram of dry material mg CAE/g dw). Some common units of measure should be included in the head of the table.

table 1- (reference 26) the polyphenolic compounds are not identified by GC-MS analysis - this must be corrected

Discussions- Lines 28- Usually, a greater number of compounds are identified in the final extract, and the final yield of the extraction increases with increasing temperature - this phrase cannot be taken into account because the number of identified compounds depends on the experimental conditions, method of identifying compounds - ex. the number of available standards, not all components are always identified. For this reason, the number of components is not a parameter to be taken into account for comparison. This appears in several sentences in this chapter.

Chapter 3.4. - does not provide sufficiently clear explanations, and the formulas are not useful for explanations.

Author Response

(The authors gave the same response as above.)

Round 2

Reviewer 1 Report

Comments and Suggestions for Authors

The manuscript has been well improved according to the suggestions of the reviewers.

However, there are some minor points which needed to be amended

The authors have numbered  phenolic compounds (1-8) and flavonoids (1-7). 

I order to avoid confusions, the numbering of flavonoids should differ from that of phenolic acids . Therefore,  could you modify this point in the text, and figure and legend for flavonoids numbering them  as 8-15

 Could you revise the style of the references. All the titles  of articles have to be witten in minuscule  letters. Actually there are not homogeneous , some of them are correct another one, not

As exemple ref 10 

Future Antimicrobials: Natural and Functionalized Phenolics. --->  Future antimicrobials: natural and functionalized phenolics. 

Etc……..

All Latin name shave to be italicized --àcheck again all the references in this regard

Author Response

Dear Reviewer,

Thank you for giving us the opportunity to submit a revised draft of our manuscript titled Using Subcritical Water to Obtain Polyphenol-Rich Extracts with Antimicrobial Properties. We have worked the previous version of the manuscript and have been able to incorporate minor changes to reflect most of the suggestions provided by the reviewers.

We have provided a point-by-point response to the comments. Please see the attachment.

We look forward to hearing from you and to responding to any further questions and comments you may have.

Sincerely,

Tjaša Žagar, Rok Frlan, Nina Kočevar Glavač

Reviewer 3 Report

Comments and Suggestions for Authors

In accordance with the reviewers' suggestions, the changes made by the authors   bring clarifications and improve the quality of the manuscript.

Author Response

Dear Reviewer,

Thank you for giving us the opportunity to submit a revised draft of our manuscript titled Using Subcritical Water to Obtain Polyphenol-Rich Extracts with Antimicrobial Properties. We appreciate the time and effort that you dedicated to providing your valuable feedback on our manuscript. We have worked the previous version of the manuscript and have been able to incorporate minor changes to reflect most of the suggestions provided by the reviewers, please find them highlighted within the manuscript.

We look forward to hearing from you and to responding to any further questions and comments you may have.

Sincerely,

Tjaša Žagar, Rok Frlan, Nina Kočevar Glavač